# Does Music Intervention Improve Anxiety in Dementia Patients? A Systematic Review and Meta-Analysis of Randomized Controlled Trials

**DOI:** 10.3390/jcm12175497

**Published:** 2023-08-24

**Authors:** Berne Ting, Daniel Tzu-Li Chen, Wei-Ti Hsu, Chih-Sung Liang, Ikbal Andrian Malau, Wei-Chih Li, Sheau-Ling Lee, Li Jingling, Kuan-Pin Su

**Affiliations:** 1Ph.D. Program for Aging, College of Medicine, China Medical University, Taichung 40402, Taiwan; berne.ting@gmail.com; 2Mind–Body Interface Laboratory (MBI-Lab), China Medical University Hospital, Taichung 40402, Taiwan; u105023415@cmu.edu.tw (D.T.-L.C.); ikbalgan@gmail.com (I.A.M.); right0455@gmail.com (W.-C.L.); 3M.D.-Ph.D. Program, College of Medicine, China Medical University, Taichung 40402, Taiwan; 4School of Chinese Medicine, College of Chinese Medicine, China Medical University, Taichung 40402, Taiwan; 5Graduate Institute of Biomedical Sciences, College of Medicine, China Medical University, Taichung 40402, Taiwan; u108305203@cmu.edu.tw; 6Department of Anesthesiology, China Medical University Hospital, Taichung 40402, Taiwan; 7Beitou Branch, Tri-Service General Hospital, National Defense Medical Center, Taipei 11490, Taiwan; lcsyfw@gmail.com; 8School of Medicine, College of Medicine, China Medical University, Taichung 40402, Taiwan; 9National Health Research Institutes, Miaoli 35053, Taiwan; sllee@nhri.edu.tw; 10An-Nan Hospital, China Medical University, Tainan 70965, Taiwan

**Keywords:** anxiety, Alzheimer’s disease, dementia, music intervention, music therapy

## Abstract

Music interventions (MIs) have been widely used to relieve anxiety in dementia in clinical settings. However, limited meta-analysis with randomized controlled trials (RCTs) on this topic has been conducted so far. A systematic search was conducted in four major databases (PubMed, EMBASE, Web of Science, and Cochrane Library) for data provided by RCTs from the inception to February 2023. The search strategy employed the terms “anxiety AND music AND dementia OR Alzheimer’s disease”. Thirteen RCTs (827 participants) were included. The results showed MI reduced anxiety significantly (SMD = −0.67, *p* < 0.001), especially for Alzheimer’s disease (*p* = 0.007) and Mixed (*p* < 0.001)-type dementia. Moreover, significant improvements in agitation (*p* = 0.021) and depression (*p* < 0.001) in dementia were observed. Additionally, several psychological mechanisms which may be associated with MI were reviewed comprehensively. In conclusion, our findings support the efficacy of MI in alleviating anxiety symptoms in dementia patients. PROSPERO Registration (ID: CRD42021276646).

## 1. Introduction

Dementia and mild cognitive impairment (MCI) are prevalent and common brain disorders among the elderly, exhibiting the involvement of multiple cognitive domains. Early intervention is recommended in the stage of MCI in case it deteriorates the progress of dementia development, and further influences patients’ daily quality of life (QoL) in more than one cognitive domain [1]. The significance of anxiety symptoms in individuals with dementia or MCI is frequently overlooked and inadequately addressed, resulting in low intervention rates and patient withdrawal due to side effects [2]. The prevalence of any anxiety disorder in the dementia population was reported to be approximately 33% [3,4], and it is often regarded as a psychological indicator of cognitive decline [2]. Considering the global prevalence of dementia, which affects over 55 million individuals, with 10 million new cases witnessed annually [5], anxiety places a burden on caregivers, healthcare providers, and financial resources [6]. 

Anxiety is a prevalent yet highly diverse condition that lacks definitive biomarkers or a singularly effective treatment [7,8]. Neglecting to address anxiety can lead to the deterioration of mental and psychological functioning, increased risk of depression, agitation, and suicide ideation and behavior [4,9,10,11]. Among patients with dementia, pharmacotherapy currently serves as the primary treatment for anxiety, though there are clinical concerns regarding the medication’s limitations and adverse effects. For instance, benzodiazepines (BZDs) may cause constipation, muscle weakness, and an increased risk of falls, while selective serotonin reuptake inhibitors (SSRIs) may result in nausea, diarrhea, headache, and insomnia [12]. In contrast to medication, non-pharmacological alternatives such as dietary modifications, physical activity, and music intervention (MI) have been reported with clinical evidence and recommended as anxiety treatments [13,14,15].

MI, encompassing both active music therapy (AMT) and passive music therapy (PMT) [16], is widely utilized in the management of anxiety disorders due to its affordability, accessibility, and lack of adverse effects [17]. Previous studies have provided evidence of the efficacy of MI in reducing anxiety symptoms among patients with dementia, including Alzheimer’s disease (AD), MCI, and vascular dementia (VaD) [18,19]. Additionally, a meta-analysis has concluded that MI can generally improve anxiety symptoms [20]. The emotional impact of music and its ability to facilitate verbal expression makes it a valuable tool in therapy [18]. Furthermore, MI has been shown to activate brain functions that play a crucial role in maintaining mood stability and reducing anxiety and depression [17,21,22].

Multiple investigations have examined the application of MI for alleviating anxiety symptoms in patients with dementia, encompassing a variety of randomized controlled trials (RCTs) and meta-analyses. Nevertheless, discrepancies in the findings of these studies have been observed [23,24,25]. Moreover, due to the scarcity of supportive high-level evidence, the objective of this study is to consolidate the existing evidence on the efficacy of MI in reducing anxiety among individuals with dementia. Consequently, we have undertaken this meta-analysis of RCTs to compare the impact of MI on anxiety symptoms in the context of dementia.

## 2. Materials and Methods

### 2.1. Protocol and Registration

The protocol for this meta-analysis was conducted following PRISMA (Preferred Reporting Items for Systematic Reviews and Meta-Analyses) [26] guidelines and was registered with the International Prospective Registry of Systematic Reviews PROSPERO (ID: CRD42021276646). Details can be found in the Appendix A.

### 2.2. Search Strategy and Selection Criteria

We searched PubMed, Embase, Web of Science, and the Cochrane Library for RCTs from the inception date of the database to February 2023. We used the strategy of “music AND anxiety AND dementia OR Alzheimer’s disease” with the Boolean search to select and filter literature in the databases. The following terms were used to guide the search strategy: PICO (Patients—Anxiety in dementia; Intervention—Music interventions; Comparison–Control, no intervention, or other non-pharmacological intervention; Outcomes—Anxiety, Depression, and Agitation assessment). We then removed the duplicates and articles that did not focus on dementia. Afterward, the titles and abstracts of all identified articles were evaluated. The inclusion criteria were as follows: (1) RCT. (2) Intervention groups that received MI that included all three factors of music (rhythm, melody, and harmony); the control group was the usual treatment or rehabilitation training. (3) Anxiety measurement was included in the outcome assessments. (4) All participants were above the age of 60 (inclusive). The exclusion criteria were as follows: (1) non-RCTs (review articles, medical protocols, conference articles, case reports, letters, editorials, pilot studies, and pilot RCT). (2) MI was administered with other therapy or was part of the complementary or alternative therapy set. (3) The control groups had any component of the accepted music, including rhythm, melody, and harmony. (4) Studies that did not provide information about the primary outcome analysis. Finally, the full text of the identified articles was evaluated for the meta-analysis.

### 2.3. Data Extraction

We developed a table to extract appropriate data, including the following details: (1) characteristics of the articles (authors, year of publication, and country); (2) the characteristics of the participants (e.g., condition and type of dementia); (3) the study design and methodological quality (random assignment, blinding, participant selection process, and follow-up status); (4) MI (method, music style, and performance style); (5) the outcome measures and statistics (type of anxiety scale, the results of the anxiety score, and other psychological scores in dementia). Three authors (Ting, Chen, and Malau) extracted data independently and resolved differences through discussions with two other authors (Li and Su).

### 2.4. Risk of Bias Assessment of Included Studies

The risk of bias in the included studies was independently assessed by two authors (Ting and Malau) according to Cochrane Collaboration’s tool [27]. The seven items used to assess quality and bias judgments were: (1) random sequence generation (selection bias), (2) allocation concealment (selection bias), (3) blinding of participants and personnel (performance bias), (4) blinding of outcome assessment (detection bias), (5) incomplete outcome data (attrition bias), (6) selective reporting (reporting bias), (7) other bias. Each item was rated as being at “low risk,” “unclear risk,” or “high risk” of bias, depending on the article. In this study, the blinding of participants and personnel was assessed separately because the administrators and investigators of the interventions may not have been the same. We decided that “selective reporting” would determine whether the clinical trial was enrolled. In addition, “other bias” would examine the details of conflicts of interest, funding sources, or the presence of more dropouts. Any disagreement with the results was resolved through discussion with the author (Chen).

### 2.5. Classification of the Music Intervention and Control Groups

We classified the MI group with live music interaction by a music therapist or professional as AMT. As PMT, pre-recorded music was played to participants via speaker (SPK). In the control group classification, any activity intervention was referred to as “active” (excluding the music factor), and no interventions, usual treatment, rest, and reading interventions were referred to as “passive”.

### 2.6. Statistical Analyses

All extracted outcome data included in this meta-analysis were continuously analyzed using standardized mean differences (SMDs) and 95% confidence intervals (CIs). The primary outcome was a post-intervention anxiety score of dementia in the MI and control groups. The last post-intervention time was chosen if the scale had multiple measurements. If more than one assessment score was available, the score provided by the objective investigator would be the primary outcome. Secondary outcomes were other psychological scores, including depression and agitation of dementia. If no standard deviation (SD) or 95% CI was reported in the original article, the effect size was estimated based on the median, interquartile range (IQR), range, standard error, t-value, or *p*-value using a random effects model [28]. In multi-arm studies, studies with more than two intervention groups were included in the meta-analysis using methods that (1) omitted groups unrelated to the comparison made and (2) included multiple groups eligible for the intervention as experimental or comparison groups to form a single paired comparison [28]. The interpretation of the effect sizes according to the Cohen guidelines is as follows: effect size = 0.2 is considered a “small” effect size, 0.5 represents a “medium” effect size, and 0.8 is a “large” effect size [29]. All *p*-values were bilateral, and 0.05 was considered statistically significant. Each heterogeneity analysis was assessed using I-square (*I*^2^) statistics. A *p*-value of less than 0.1 for the *I*^2^ test indicates significant heterogeneity [28]. Potential publication bias was calculated via funnel plot and Egger regression asymmetry analysis [30]. Two authors performed data analysis (Ting and Hsu) and two authors disputed resolution (Li and Chen).

### 2.7. Subgroup Outcomes

We found that the studies included in our research exhibited distinct classifications, pertaining to AD or a combination of multiple conditions (Mixed). Therefore, we further divided them into two different dementia groups: AD and the Mixed group. On the other hand, for music-integrated therapy, we split the studies into two types: AMT and PMT. Five subgroups were selected for music types: (1) Improvisation: live, real-time performances for the participants, with interactivity. (2) Participant preferences: participants selected their preferred music or song before the test. (3) Multiple music: combinations of two or more types of music. (4) Old song: a kind of popular music from an older era, as time has passed, and music exists in the memories of the people from this period. (5) Instrumental music: music without any vocals or lyrics. Different countries or regions also have various instruments and playing characteristics. As for the presentation type, two subgroups were created: live music and pre-recorded. Live music refers to the interaction of participants with a music therapist playing an instrument, while SPK refers to pre-recorded music played through speakers. The rating scale used to assess anxiety in dementia was mainly based on Rating Anxiety in Dementia (RAID), while if other rating tools such as the Hamilton Anxiety Scale (HAM-A), State Trait Anxiety Inventory (STAI), or Neuropsychiatric Inventory (NPI) were used, they would be grouped as non-RAID. We performed subgroup analyses to investigate the potential heterogeneity of the included studies. We used comprehensive meta-analysis software version 3 (Biostat, Englewood, NJ, USA) to process the statistics of all included studies.

## 3. Results

### 3.1. Identification of Eligible Studies

Figure 1 presents the flow chart and outcomes of our screening process. Our search strategy yielded a total of 654 articles, of which 430 duplicate articles were excluded. Subsequently, we conducted a thorough evaluation of the titles and abstracts of the remaining 224 articles. Following the application of our screening criteria, 184 articles were excluded, leaving us with 40 full-text articles for further analysis. From this set, 27 articles were excluded as they did not meet the criteria for being non-randomized controlled trials, pilot studies, protocols, reviews, or relevant outcomes. Ultimately, our meta-analysis included a total of 15 trials from 13 articles. Further information regarding the 27 excluded articles can be found in Appendix A.

### 3.2. Study Characteristics and Patient Population

A total of 827 participants were included in 13 articles published between 2009 and 2022 (Table 1). The sample size for each article ranged from 18 to 112 individuals, aged from 65 to over 95 years old. As for the dementia types, four were AD, and the others were Mixed type. Based on the definition of the Mini-Mental Status Exam (MMSE) or Montreal Cognitive Assessment (MoCA), the severity of the cognitive impairment was as follows: four were mild, one was mild-to-moderate, seven were moderate, and one exhibited severe cognitive impairment. As for the MI type used in the included studies, seven studies used AMT, three used PMT, and three used a combination of AMT and PMT. Regarding the music style, two studies used improvised music, three used multiple kinds of music, four used old songs, and four used music chosen by the participants based on their preferences. The music was presented in the form of live music performances in seven studies, with pre-recorded music played through SPK in five studies. The control types were ‘Active’ in seven studies, ‘Passive’ in five studies, and waitlist in one study. The frequency of interventions MI in the included studies ranged from 20 to 120 min, one to five times per week, with a minimum of one week and a maximum of 24 weeks. Regarding the rating scales for anxiety, six were RAID, and seven were non-RAID, including three HAM-A, two NPI, and two STAI. Two articles with double interventions met the inclusion criteria; therefore, multiple comparisons were performed. Eventually, 15 tails from 13 articles (MI and control group) were extracted for meta-analysis. We describe the detailed classification of the control group, and the content was categorized based on whether the patient had received non-music intervention or received the usual treatment (Table 1).

### 3.3. Quality Assessment of Included Articles

Among all included articles, two were at high risk of selection bias [31,32] and one was at high risk of personnel unevenness [42]. Five of the included articles were at high risk of performance bias due to the nature of MI [31,32,36,42,43]. Three articles had a high risk of bias in blinding outcome assessment [31,42,43]. As for detection bias, all articles were at low risk due to the objective assessment scales used in the study (Appendix A).

### 3.4. Outcomes of Music Intervention

Figure 2 shows the primary outcome of our extracted data (*k* = 15) of 13 RCTs involving 827 participants. We found that MI significantly reduced anxiety scores for dementia (SMD = −0.67, 95% CI = −0.94 to −0.40, *p* < 0.001), with significant heterogeneity (*I*^2^ = 70.26%, *p* for *I*^2^ < 0.001). We then conducted a subgroup analysis, examining each subgroup’s heterogeneity to confirm the study’s stability.

### 3.5. Subgroup Analysis

The subgroup analysis (Table 2) shows the effect of MI on anxiety in dementia in the actual data. In general, the impact of MI on anxiety reduction for dementia types was shown significantly in both AD (SMD = −1.15, *p* = 0.007) and Mixed (SMD = −0.49, *p* < 0.001). Both AMT (SMD = −0.80, *p* = 0.001) and PMT (SMD = −0.97, *p* = 0.002) were significant in the MI treatment category. And the combination of AMT and PMT (SMD = −0.24, *p* = 0.073) was insignificant. Among the music type, patient preference (SMD = −0.35, *p* = 0.005) and multiple music (SMD = −0.98, *p* = 0.005) were the most effective types, and old song (SMD = −0.79, *p* = 0.017) also significantly improved anxiety in dementia; improvisation (SMD = −0.61, *p* = 0.247) did not significantly improve anxiety in dementia. In terms of music performance scores, both live music (SMD = −0.651, *p* = 0.001) and pre-recorded music played through SPK could improve anxiety symptoms (SMD = −0.70, *p* = 0.001). Both RAID (SMD = −0.36, *p* < 0.001) and non-RAID (SMD = −1.08, *p* < 0.001) showed no differences in anxiety-accessing scores.

### 3.6. Secondary Outcome

Figure 3 shows the forest plots for the secondary outcomes of agitation and depression in dementia, assessed according to the Cohen-Mansfield Agitation Inventory (CMAI), a tool that is widely used to investigate neuropsychiatric inventory (NPI). A total of 248 participants and 6 data were included for analyses. MI could significantly improve agitation in dementia (SMD = −0.40, 95% CI = −0.74 to −0.06, *p* = 0.021) (Figure 3a). However, moderate heterogeneity still existed (*I*^2^ = 40.98%, *p* for *I*^2^ = 0.132). For depression in dementia, with 588 participants and 10 data analyses, the score showed statistical significance after MI (SMD = −0.45, 95% CI = −0.68 to 0.22, *p* < 0.001). Also, the data has moderate heterogeneity (*I*^2^ = 46.65%, *p* for *I*^2^ < 0.051) (Figure 3b).

### 3.7. Evaluation of Publication Bias

Funnel plots for publication bias tests were conducted for all studies performed using CMA 3.0 software (Figure 4). Most studies were clustered at the right side of the effect size, while the left side was more dispersed. The majority of the studies were concentrated inside the diagonal line. This funnel plot features a study of MI for anxiety in dementia, which was examined using the “Trim and Fill” [44] and Eggers tests [30] to verify the publication bias. Egger’s test showed no significant publication bias in this research (intercept = −2.29, 95% CI = −5.84 to 1.26, *t* = 1.398, *df* = 13, 1st-tailed *p* = 0.093, 2nd-tailed *p* = 0.186).

## 4. Discussion

To the best of our knowledge, this study represents the initial exploration into the efficacy of MI in addressing anxiety as the primary concern in individuals with dementia. Our first main finding demonstrates a significant reduction in anxiety symptoms among patients with various types of dementia following MI intervention. To be specific, this meta-analysis of 13 RCTs (*n* = 827) reveals a highly favorable impact of MI on anxiety reduction in dementia patients (SMD = −0.67, *p* < 0.001). A recent review article has concluded that the effectiveness of MI may vary depending on the type of dementia, with greater effectiveness observed in Mixed type compared to AD due to the progressive decline in memory and cognitive function [45]. Importantly, our results are consistent with previous findings [46,47,48], providing the highest level of evidence. Moreover, other clinical trials have reported similar positive effects of MI in treating anxiety and other symptoms in patients with neurodegenerative or psychiatric disorders [23,25]. MI has been found to yield multiple positive effects from several perspectives. For instance, it has been widely reviewed for its psychological and mental health benefits, including the reduction of depression, anxiety, and stress symptoms, as well as improvement of mood, cognitive function, and relaxation [49,50]. Additionally, clinical assessments have shown benefits such as decreased heart rate, blood pressure, cortisol levels, and increased levels of endorphins and immune function [51,52].

The effectiveness of MI may be attributed to various mechanisms. Firstly, MI can regulate mood by stabilizing the autonomic nervous system (ANS), leading to heightened arousal in individuals with anxiety and depression [53,54]. MI employs multisensory and active approaches to target the ANS and enhance mood regulation [16,55]. Additionally, previous studies have indicated that music can stimulate the release of neurotransmitters, such as dopamine and serotonin, which contribute to mood regulation and anxiety reduction. This can further improve cognitive, emotional, and motor function in individuals with neurological conditions [56]. Secondly, MI is associated with the activity of the hypothalamic–pituitary–adrenal (HPA) axis, and further reduces cortisol levels, thereby alleviating stress, anxiety, and depression [57,58]. The HPA axis plays a crucial role in the stress response, in which patients with psychiatric disorders may be hyperactive. Similar to its regulation of the ANS, MI promotes feelings of relaxation and stimulates the release of dopamine and serotonin, which can reduce inflammation and positively impact brain function. This ultimately helps regulate mood and reduce HPA activation [16]. Moreover, MI can serve as a distraction or coping strategy, assisting individuals in managing stress and negative emotions, thereby reducing HPA axis activation. Thirdly, MI may improve neuroplasticity, further improving cognitive function and behavioral symptoms [59]. Neuroplasticity refers to the brain’s ability to adapt to different situations, and increasing its level may help improve psychiatric symptoms such as depression and anxiety [60]. MI can affect multiple brain regions and facilitates the creation of new neural connections through various sensory modalities. For instance, previous studies have shown significant increases in functional connectivity between specific brain regions after patients received MI, including the bilateral dorsolateral prefrontal cortex and frontal areas [61], the dorsal anterior insula and posterior insular networks [62], and the right middle temporal gyrus [63]. Furthermore, functional magnetic resonance imaging (fMRI) [64] and near-infrared spectroscopy (NIRS) [65] studies have also demonstrated increased activity in specific brain regions after receiving MI. Therefore, interventions like MI that enhance neuroplasticity are recommended for anxiety patients in clinical settings [66]. Lastly, MI may have anti-inflammatory effects on the brain and therefore improves psychiatric symptoms. Emerging evidence suggests that neuroinflammation is key to the psychopathology of neurodegenerating disorders such as AD and psychiatric symptoms such as anxiety [67]. Various anti-inflammatory interventions, including Omega-3 fatty acid [14], yoga [68], Tai-Chi [69], and acupuncture [70], have been widely accepted and applied in clinical practice. MI, specifically, can modulate the immune response by activating the vagus nerve, releasing anti-inflammatory cytokines, and reducing pro-inflammatory cytokines. For instance, a significantly lower level of plasma interleukin-6 (IL-6) was observed in the group that received MI compared to the control group [71]. Additionally, MI can reduce stress and promote relaxation, which decreases the production of stress hormones like cortisol which increase brain inflammation [72]. Thus, MI can help alleviate anxiety in dementia patients through various mechanisms, including regulation of the ANS, stabilization of the HPA axis, improvement of neuroplasticity, and inhibition of neuroinflammatory status.

Our second main finding indicates that anxiety symptoms can be alleviated through either AMT or PMT types of MI. Previous research has demonstrated the effectiveness of both AMT and PMT in promoting mental health, with the choice depending on individual needs, preferences, and abilities [16,73]. The critical distinction between AMT and PMT is that AMT involves active engagement in music making to express emotions and enhance social skills [74], whereas PMT employs pre-recorded music to elicit emotional responses [75]. In addition to the aforementioned psychological and clinical advantages, MI has been widely associated with psychosocial benefits. For instance, both AMT and PMT have been shown to enhance communication and social interaction skills, as well as boosting self-esteem, emotional expression, and social support [76]. Specifically, the rhythm and melody of music stimulate calming effects, reducing stress levels and promoting relaxation, thereby aiding individuals with dementia in experiencing less anxiety [77,78]. AMT may facilitate self-expression, emotional understanding, coping skill development, emotional regulation, and improvement in social skills and relationships for dementia patients [74,79,80]. By contrast, PMT does not require active participation in music activities. Nevertheless, PMT help manage anxiety symptoms by providing calming and soothing effects, promoting relaxation, and facilitating emotional regulation through reflection [25,81,82]. In summary, AMT and PMT are beneficial for anxiety in dementia patients through various psychological mechanisms, including emotional regulation and relaxation, enabling self-expression, improvement in social skills, and a reduction in stress levels.

Our third finding is that significant anxiety relief effects were shown in MI with the music types of multiple music, old song, and patients’ preferences, while the same was not true for improvisation. This interesting result is partially consistent with the previous study. Sittler et al. reported that MI can improve the emotional well-being of patients with dementia, especially when tailored to their music preferences. In clinical settings, the choice of music in MI can be customized based on individual needs and preferences. The flexibility of MI may be crucial to the effectiveness of music therapy in reducing anxiety in dementia [42,58,83,84]. On the other hand, the use of improvisation in MI did not yield a significant effect in our study, possibly due to the relatively small sample size (*k* = 2). However, it is worth noting that some studies have emphasized the importance of applying appropriate improvisation techniques, and therapists should be aware of its limitations [85,86]. Further investigation focusing on improvisation is warranted for future research.

Our subgroup analysis showed that significance was observed in both the RAID and non-RAID groups. To account for the potential influence of the rating scale on the results, we assessed the heterogeneity between each group, and no significant heterogeneity (*k* = 8, *p* = 0.730) was found. The RAID scale demonstrates good face validity, expert judgment, and common-sense support as an anxiety measure in dementia [87]. Furthermore, it exhibits good convergent validity, as it correlates well with other anxiety measures such as the Geriatric Anxiety Inventory and the Hospital Anxiety and Depression Scale-Anxiety. Higher scores on the RAID scale are associated with increased levels of behavioral and psychological symptoms of dementia and decreased QoL. In general, the RAID scale is a valid measure of anxiety in dementia, supported by its face, convergent, and criterion validity [88]. Alternative anxiety scales, such as the Hamilton Anxiety Rating Scale (HAM-A), the Neuropsychiatric Inventory (NPI), and the State-Trait Anxiety Inventory (STAI), have been utilized in other studies, in contrast to the non-RAID anxiety scales used in this study. While the HAM-A is widely used and possesses good psychometric properties, it is not specifically designed for individuals with dementia and may not effectively differentiate between anxiety and depression symptoms [89]. The NPI evaluates behavioral symptoms but lacks a specific and refined measure of anxiety [90]. The STAI, while also widely used with good psychometric properties, may be susceptible to response bias and unsuitable for individuals with severe cognitive impairments [91]. In our analysis, we observed high heterogeneity in these scales (*k* = 7, I^2^ = 76.86%, *p* < 0.001), but this did not impact the collated analysis. In addition, other important and interesting findings from our secondary outcome are that significant improvements were reported in both depression and agitation in dementia. Depression often coexists with anxiety in clinical settings [92], and previous research has explored the potential benefits of using MI to alleviate depressive and anxious symptoms in individuals with dementia [93,94,95,96]. On the other hand, our finding regarding agitation aligns with the results of a previous meta-analysis [93]. It has been suggested that these effects may be attributed to PMT, which involves singing and listening to music to alleviate symptoms. However, the study’s sample size exploring this effect was limited, and further research is necessary to extend and expand upon these findings.

### Strength and Limitation

This study is characterized by two key features. Firstly, a comparative analysis was conducted using data collected from the baseline and post-intervention phases, which allowed for a more precise estimation and assessment of the study’s objectives. Secondly, the sample size of the included studies was significantly larger than previous meta-analyses that focused on patients with dementia [25,97,98]. These advantages are expected to enhance the accuracy of the results. However, this study had several limitations. Firstly, the inclusion of additional keywords in the search strategy, such as “singing” or “move rhythm,” may potentially allow more studies to be included. Secondly, future investigations could consider conducting more specific analyses of the MI function, despite the Mixed major dementia type. Moreover, length and frequency of the MI should be considered and analyzed. Thirdly, the identification of double-blind RCTs is relatively challenging due to the characteristics of MI, which may indicate a higher risk. Fourthly, although efforts were made to address the high heterogeneity in the subgroup analysis, there was still considerable variation, suggesting the need for further exploration of other factors. Lastly, the exclusion of active control, i.e., control groups that involve any component of the accepted music, including rhythm, melody, and harmony, may have led to overestimation of the effectiveness of MI. A more comprehensive study design, including active control, should be considered in future investigation.

## 5. Conclusions

This study provides supportive evidence that MI is an effective strategy for reducing anxiety symptoms in dementia patients. The consensus of practice guidelines and standardized training of MI needs to be established to translate the research findings into clinical applications for anxiety in dementia patients in clinical settings.

## Figures and Tables

**Figure 1 jcm-12-05497-f001:**
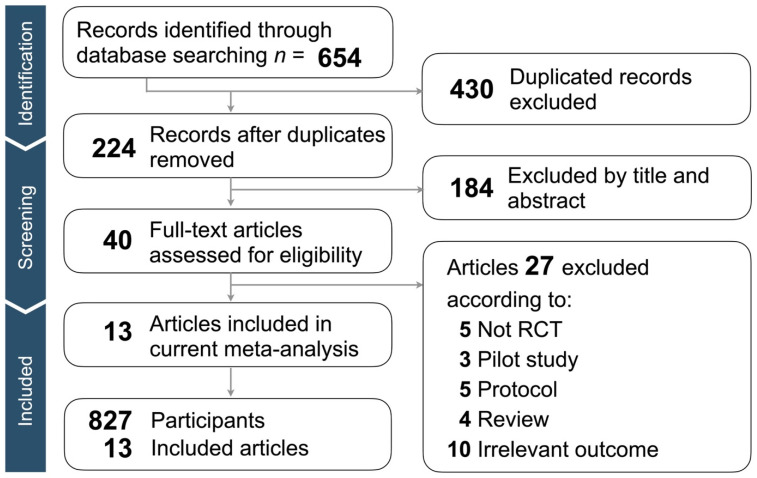
Flow chart of the selection strategy and inclusion and exclusion criteria.

**Figure 2 jcm-12-05497-f002:**
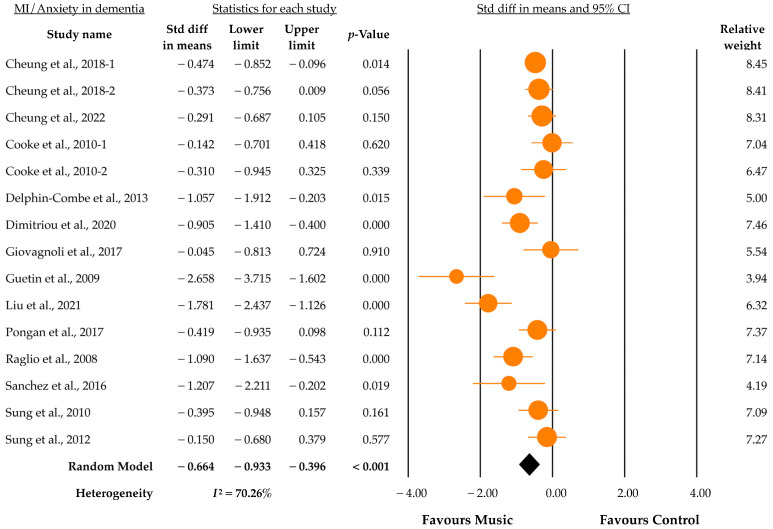
Forest plot of the effect of music on anxiety in dementia scores [31,32,33,34,35,36,37,38,39,40,41,42,43].

**Figure 3 jcm-12-05497-f003:**
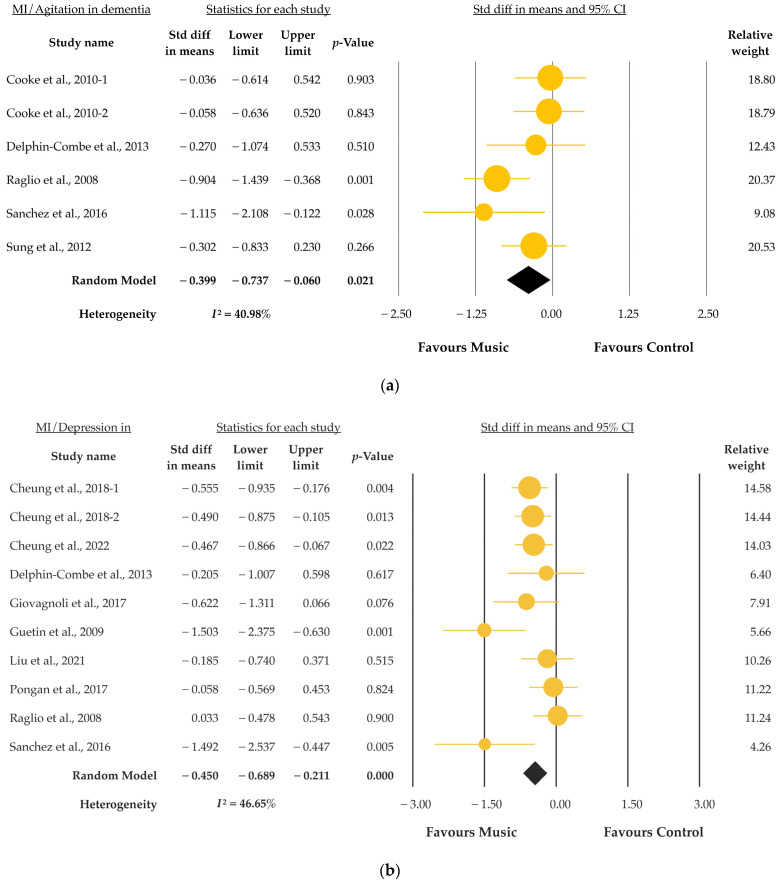
(**a**) Forest plot of MI to agitation in dementia; (**b**) Forest plot of MI to depression in dementia [31,32,33,34,36,37,38,39,40,41,43].

**Figure 4 jcm-12-05497-f004:**
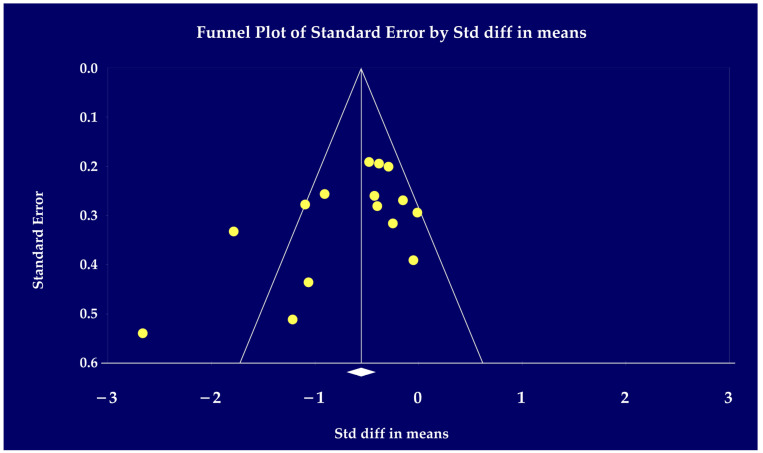
Funnel plot of standard difference means on the *X*-axis and standard error on the *Y*-axis for the effect of music on anxiety in dementia release [31,32,33,34,35,36,37,38,39,40,41,42,43].

**Table 1 jcm-12-05497-t001:** General Characteristics of Music Intervention Studies on anxiety in dementia.

Authors and Year	Country	Study Design	Comparison	N	Age, Y Mean (SD)	Dementia Types	Dementia Severity Level	Session Style	Type/Genres/Titles of Music	Equip	Control Type	Control Descriptions	Frequency	Rating Scales
Cheung et al., 2018 [31]	Hong Kong	Multi-RCT	Music-1 Music-2 Social activity	58 54 53	85.71 (6.68) 84.50 (6.82) 85.58 (7.46)	Mixed	Moderate	AMT/PMT	Multiple music	Live/SPK	Active	Social activity	40 min/twice a week/6 weeks	RAID
Cheung et al., 2022 [32]	Hong Kong	Cluster-RCT	Music Control	55 45	79.53 (8.53)	Mixed	Moderate	AMT	Patients Preferences	SPK	Waitlist	Waitlist	30–45 min/3 times a week/12 weeks	RAID
Cooke et al., 2010 [33]	Australia	Crossover	Music Control	23 23	65–95+	Mixed	Moderate	AMT/PMT	Patients Preferences	Live	Passive	Reading	40 min/3 times a week/8 weeks	RAID
Delphin-Combe et al., 2013 [34]	France	RCT	Music Control	12 12	79.20 (6.90) 79.00 (6.70)	AD	Moderate	AMT	Multiple music	Live	Active	Board games	30 min/5 times a week/2 weeks	HAM-A
Dimitriou et al., 2020 [35]	Greece	Multi-Crossover	Music Exercise	20 20	74.70 (7.17)	Mixed	Mild	PMT	Old songs	SPK	Active	Exercise/Aromatherapy and Massage	45 min/5 times a week/1 week	NPI
Giovagnoli et al., 2017 [36]	Italy	Multi-RCT	Music Cognitive training	17 17	73.92 (7.74) 73.50 (5.96)	Mixed	Moderate	AMT	Improvisation	Live	Active	Cognitive training	45 min/twice a week/12 weeks	STAI
Guétin et al., 2009 [37]	France	RCT	Music Control	14 12	85.20 (6.00) 86.90 (5.20)	AD	Mild	PMT	Multiple music	Live	Passive	Rest and reading	20 min/once a week/24 weeks	HAM-A
Liu et al., 2021 [38]	Taiwan	RCT	Music Control	25 25	86.60 (4.50) 86.90 (5.70)	AD	Mild	AMT	Old songs	Live	Passive	Rest and reading	60 min/once a week/12 weeks	HAM-A
Pongan et al., 2017 [39]	France	RCT	Music Control	31 28	78.80 (7.43) 80.20 (5.71)	AD	Mild	AMT	Patients Preferences	Live	Active	Painting	120 min/once a week/12 weeks	STAI
Raglio et al., 2008 [40]	Italy	RCT	Music Control	30 29	84.40 (5.50) 85.80 (5.40)	Mixed	Moderate	AMT	Improvisation	Live	Active	Educational and entertainment activities	30 min/10 times/16 weeks	NPI
Sánchez et al., 2016 [41]	Spain	RCT	Music Control	99	88.73 (7.36) 88.09 (6.80)	Mixed	Sever	PMT	Patients Preferences	SPK	Active	Multi-Sensory Stimulation	30 min/twice a week/16 weeks	RAID
Sung et al., 2010 [42]	Taiwan	RCT	Music Control	29 23	80.12 (7.55)	Mixed	Moderate	AMT	Old songs	SPK	Passive	Treatment as usual	30 min/twice a week/6 weeks	RAID
Sung et al., 2012 [43]	Taiwan	RCT	Music Control	27 28	81.37 (9.14) 79.50 (8.76)	Mixed	Mild-Moderate	AMT/PMT	Old songs	SPK	Passive	Treatment as usual	30 min/twice a week/6 weeks	RAID

Abbreviation: AD: Alzheimer’s Disease; Mixed: mixed types of dementia; RAID: Rating Anxiety in Dementia; STAI: State-Trait Anxiety Inventory; NPI: neuropsychiatric Inventory; Live: live music; SPK: speaker; AMT: active music therapy; PMT: passive music therapy; AMT/PMT: active and passive.

**Table 2 jcm-12-05497-t002:** Subgroup analysis MI of anxiety in dementia.

Subgroup	*k*	Effect Size (SMD)	95% Confidence Interval	*p*
**Dementia type**				
AD	5	−1.144	−1.978 to −0.310	0.007
Mix	10	−0.481	−0.681 to −0.281	<0.001
**Therapy type**				
AMT	6	−0.798	−1.256 to −0.340	0.001
PMT	5	−0.967	−1.566 to −0.367	0.002
AMT + PMT	4	−0.231	−0.484 to 0.022	0.073
**Music type**				
Improvisation	2	−0.603	−1.625 to 0.418	0.247
Multiple music	4	−0.976	−1.664 to −0.287	0.005
Old song	4	−0.789	−1.440 to −0.139	0.017
Patients’ preferences	5	−0.348	−0.592 to −0.104	0.005
**Presentation**				
Live music	8	−0.651	−1.026 to −0.276	0.001
Pre-record	7	−0.694	−1.112 to −0.276	0.001
**Rating scale**				
RAID	8	−0.355	−0.527 to −0.183	<0.001
non-RAID	7	−1.075	−1.593 to −0.557	<0.001

Abbreviation/SMD: Standardized mean difference; k: number studies; AD: Alzheimer’s Disease; Mix: mixed types of dementia; AMT: active music therapy; PMT: passive music therapy; RAID: Rating Anxiety in Dementia.

## Data Availability

Not applicable.

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
