# Peer review of "Does Music Intervention Improve Anxiety in Dementia Patients? A Systematic Review and Meta-Analysis of Randomized Controlled Trials"

_jcm, 2023, doi:10.3390/jcm12175497_

Round 1
Reviewer 1 Report
It is recommended to revise the whole manuscript regarding the English language. The topic is very interesting and this manuscript would be a great addition to the Music Therapy and Music Medicine field. However, it needs to be revised.
Major revision of the English language
Author Response
Thank you for the recommendation. We have checked and rephrased the whole manuscript regarding the language issue, especially the introduction, results, and discussion section. Please see the newly uploaded manuscript.Reviewer 2 Report
This meta-analysis will be able to serve as a valuable tool in synthesizing existing evidence, providing the impact of MI in anxiety and depression in patients with dementia. This knowledge can inform clinical practice, improve patient care, and guide future research efforts in this important area.
I would like to offer a few minor suggestions, and kindly request your attention to the details provided below for more specific information.
Line 22 “However, there has been no meta-analysis with RCTs on this….”: As there are existing meta-analyses regarding depression/neuropsychiatric symptoms in AD/MCI, I recommend toning down the statement or specifying the gap/uniqueness of this meta-analysis paper:
Wong, T. C. Effectiveness of music intervention on cognitive function and neuropsychiatric symptoms in the elderly with dementia: a meta-analysis. Frontiers of Nursing, 9(2), 143-153.
Li, H. C., Wang, H. H., Lu, C. Y., Chen, T. B., Lin, Y. H., & Lee, I. (2019). The effect of music therapy on reducing depression in people with dementia: A systematic review and meta-analysis. Geriatric Nursing, 40(5), 510-516.
Dorris, J. L., Neely, S., Terhorst, L., VonVille, H. M., & Rodakowski, J. (2021). Effects of music participation for mild cognitive impairment and dementia: A systematic review and meta‐analysis. Journal of the American Geriatrics Society, 69(9), 2659-2667.
Tang, Q., Huang, Z., Zhou, H., & Ye, P. (2020). Effects of music therapy on depression: A meta-analysis of randomized controlled trials. PloS one, 15(11), e0240862.
Line 92-93 ”(3) the control groups have any component…..and harmony.”: Could you please explain why you excluded the active control group (including musical components)?
Line 148 “We found that the included study…..”: Could you please rephrase this sentence? Did you mean that the included studies had AD or mixed dementia types?
Line 183 “…severe cognitive impairment: was it based on MoCA scores? Could you please specify it?
Section 3.6 (secondly outcome): There are typo in the section title (secondly -> Secondary)
Section 3.6: Could you please specify the assessment for the agitation and depression?
Page 11, line 59 “To the best of our knowledge,,,,,, “: Similar to my first comment, I would suggest toning down as there are existed systematic review on this topic.
Discussion section: There are some words I'd like to tone down.
ú Line 82: regulates
ú Line 90: enhances
ú Line 94: stimulates
ú Line 106: reduces
ú Line 132: helps
ú Line 138: reduce
Line 190 “…..previous meta-analyses.”: Could you please provide the references for previous meta-analyses?
For the future consideration, From a clinical perspective, it would also be helpful to know the length and frequency of the relevant music therapy/music-based intervention sessions. Could you please consider including this information as well?
Reviewer 3 Report
This study investigated the improvement of anxiety induced by dementia by Music intervention. Some minor points arise from this study are recommended:
The discussion section needs more explanations and integration.
The conclusion needs revision and adding some more descriptions.
The English language needs more revision.
Author Response
Thank you for the recommendation. We have checked and rephrased the whole manuscript regarding the language issue, and added several additional explanations and integration in the discussion section. Please find it in the uploaded attachment.